# Comparing Performances of CNN, BP, and SVM Algorithms for Differentiating Sweet Pepper Parts for Harvest Automation

**Bongki Lee [1], Donghwan Kam [1], Yongjin Cho [2,3,*], Dae-Cheol Kim [2,3] and Dong-Hoon Lee [4,*]**

1. Institute of Biotechnology and Bioengineering, Sungkyunkwan University, Suwon-si 16419, Korea; dkways@skku.edu (B.L.); kamdh@skku.edu (D.K.)
2. Department of Bio-Industrial Machinery Engineering, Jeonbuk National University, Jeonju 54896, Korea; dckim12@jbnu.ac.kr
3. Institute for Agricultural Machinery & ICT Convergence, Jeonbuk National University, Jeonju 54896, Korea
4. Department of Biosystems Engineering, Chungbuk National University, Cheongju 28644, Korea
* Correspondence: choyj@jbnu.ac.kr (Y.C.); leedh@cbnu.ac.kr (D.-H.L.); Tel.: +82-63-270-2615 (Y.C.)

**Abstract:** For harvest automation of sweet pepper, image recognition algorithms for differentiating each part of a sweet pepper plant were developed and performances of these algorithms were compared. An imaging system consisting of two cameras and six halogen lamps was built for sweet pepper image acquisition. For image analysis using the normalized difference vegetation index (NDVI), a band-pass filter in the range of 435 to 950 nm with a broad spectrum from visible light to infrared was used. K-means clustering and morphological skeletonization were used to classify sweet pepper parts to which the NDVI was applied. Scale-invariant feature transform (SIFT) and speeded-up robust features (SURFs) were used to figure out local features. Classification performances of a support vector machine (SVM) using the radial basis function kernel and backpropagation (BP) algorithm were compared to classify local SURFs of fruits, nodes, leaves, and suckers. Accuracies of the BP algorithm and the SVM for classifying local features were 95.96 and 63.75%, respectively. When the BP algorithm was used for classification of plant parts, the recognition success rate was 94.44% for fruits, 84.73% for nodes, 69.97% for leaves, and 84.34% for suckers. When CNN was used for classifying plant parts, the recognition success rate was 99.50% for fruits, 87.75% for nodes, 90.50% for leaves, and 87.25% for suckers.

**Keywords:** NDVI; image processing; SURF; SIFT; SVM; BP algorithm; performance; sweet pepper; deep neural network

## 1. Introduction

In South Korea, the need for automation of physically demanding agricultural labor is increasing because of the aging agricultural workforce and the growing proportion of women among the agricultural workforce [1]. In agricultural work such as weeding and pruning, the technology to distinguish and classify plant parts is necessary for automation of various agricultural tasks. Classification of plant parts can be performed using an image recognition technology. Once an image of a recognition target is obtained, pre-processing to extract the interested region is performed. The process of recognizing an object and transforming it into a suitable format to be processed with a computer is then performed by dividing the features of the recognition target within the area of interest [2].

Machine vision is a useful tool for plant recognition and identification [3]. To extract leaves and canopies, various algorithms such as Active Shape Model [4], Color Segmentation [5] using an RGB-D (Red, Green, Blue Depth) Camera, Support Vector Machines [6], Clustering Algorithm [7], Watershed Algorithm [8–10], and genetic algorithm [11] have been proposed and successfully used to recognize plant parts in the image. Machine recognition has been effectively used to analyze the shape and growth status of crops in

fields such as plant phenomics. PhenoAIxpert (LemnaTec, Aachen, Germany) is using machine recognition technology to acquire plant phenotypic information in an image-spectral acquisition chamber. In addition, PlantScreen (Photon Systems Instruments, Drásov, Czech Republic) acquires an image and spectral information of plants and uses machine recognition technology to select excellent plants based on plant shape and spectral response information. However, plant parts recognized via images have been acquired in a controlled experimental environment rather than in the field where plants are actually grown. In the actual farm field, the canopy, including leaves, does not exist alone. A group of plants can form a colony to affect image recognition, making it difficult to clearly differentiate plant parts [12].

Various methods such as vegetation index [13] have been used to measure crop productivity on a farm and to diagnose crop growth. Silva et al. (2016) conducted an experiment to confirm the water stress of soybeans through NDVI (Normalized Difference Vegetation Index) images and confirmed a significant correlation between water stress and the NDVI of soybeans from each experiment [14]. To introduce plant shape recognition technology through images, it is necessary to distinguish targeted plant parts from the surrounding background. In a real farm field, it is easy to misread the contrast of an object because of lighting, sunlight, and shadow, which significantly lower the recognition rate of an object [15]. To improve the recognition rate of an object in an image, various methods such as Bayesian-classifier [16] and Clustering [17] using machine learning have been proposed. Methods for improving the recognition rate of objects in images have been evaluated for their accuracies in a competition called ILSVRC (ImageNet Large Scale Visual Recognition Challenge) [18]. Prior to 2012, when machine learning was predominant, the average recognition rate was below 75%. In 2012, Krizhevsky et al. [19] proposed a convolutional neural network (CNN), an artificial neural network model, and showed a recognition rate of 85%, exceeding the existing recognition rate. Since 2012, various artificial neural network models have been proposed, showing recognition accuracy exceeding 95%, which is comparable to human recognition accuracy [20]. An artificial neural network is a pattern recognition method inspired by the interconnection of neurons in the human nervous system. A general artificial neural network consists of an input layer, a hidden layer, and an output layer. Each layer includes neurons [21]. Neurons between neighboring layers are connected by weights. Input values are learned by changing weights through repeated learning. The larger the number of hidden layers, the more complex the data that can be modeled. When the number of hidden layers is two or more, it is called a deep neural network [22]. It is necessary to apply an artificial neural network model to recognize the shape of a plant through an image. Pound et al. [23] proposed a method of classifying wheat grains, nodes, and leaves using a CNN (Convolution Neural Network) but analyzed wheat as a learning factor in a controlled indoor environment. To recognize the shape of plants in an environment where actual plants are grown, color-based 3-dimensional fruit recognition [24], fruit recognition using an LED reflected light and color model [25], color information obtained from an RGB-D camera, and recognition of the fruit stalk by classifying the surface normal vector and curvature into Support Vector Machines [26] have been reported. However, the shape of the plant based on color information could not be recognized because nodes and leaves had the same color information.

Therefore, the purpose of this study was to develop an image processing algorithm that classifies plant parts, and the target of classification was paprika, which has various colors of fruits such as red, yellow, orange, and green. Sweet pepper is one of the most cultivated greenhouse crops in South Korea [27]. This study aimed to develop image processing algorithms to classify plant parts for automation of sweet pepper farming tasks such as weeding, pruning, and fruit thinning. Another aim of this study was to evaluate performances of these algorithms. Detailed study objectives are as follows:

1. To recognize the shape of sweet pepper in a farm environment, classify each part, and develop an image processing algorithm to classify sweet pepper parts for Plenty

(red sweet pepper), President (yellow sweet pepper), and Derby (orange sweet pepper) varieties.

2.  To classify sweet pepper parts with the NDVI (Normalized Difference Vegetation Index) by dividing the targeted area using k-means clustering and morphological skeletonization followed by extraction of local features using SIFT (Scale-Invariant Feature Transform) and SURF (Speeded-Up Robust Features).
3.  To evaluate performances of developed algorithms such as the BP (Backpropagation) algorithm and the SVM (Support Vector Machine) algorithm for classifying each part (leaf, node, stem, and fruit) compared to a deep neural network algorithm.

## 2. Materials and Methods

### 2.1. Hardware Composition

The sweet pepper greenhouse maintained a temperature between 20 and 30 °C and a relative humidity between 60 and 80% [28]. A Blackfly BFLY-U3-13S2C-C color camera (Teledyne FLIR LLC, Wilsonville, OR, USA) was selected to obtain images in the sweet pepper greenhouse (Table 1). The operating temperature range of the selected camera was 0–45 °C. The humidity was 20–80%. Most images were in the 400–950 nm range to analyze the NDVI. The image sensor used in this study acquired images from 400 to 950 nm (Figure 1).

**Table 1.** Specifications of the Blackfly BFLY-U3-13S2C-CS color camera.

| Model | Imaging Sensor | Resolution | Operating Temperature | Operating Humidity |
|---|---|---|---|---|
| Blackfly BFLY-U3-13S2C-CS | Sony ICX445, 1/3″, 3.75 μm | 1288 × 964 (30 FPS) | 0–45 °C | 20–80% |

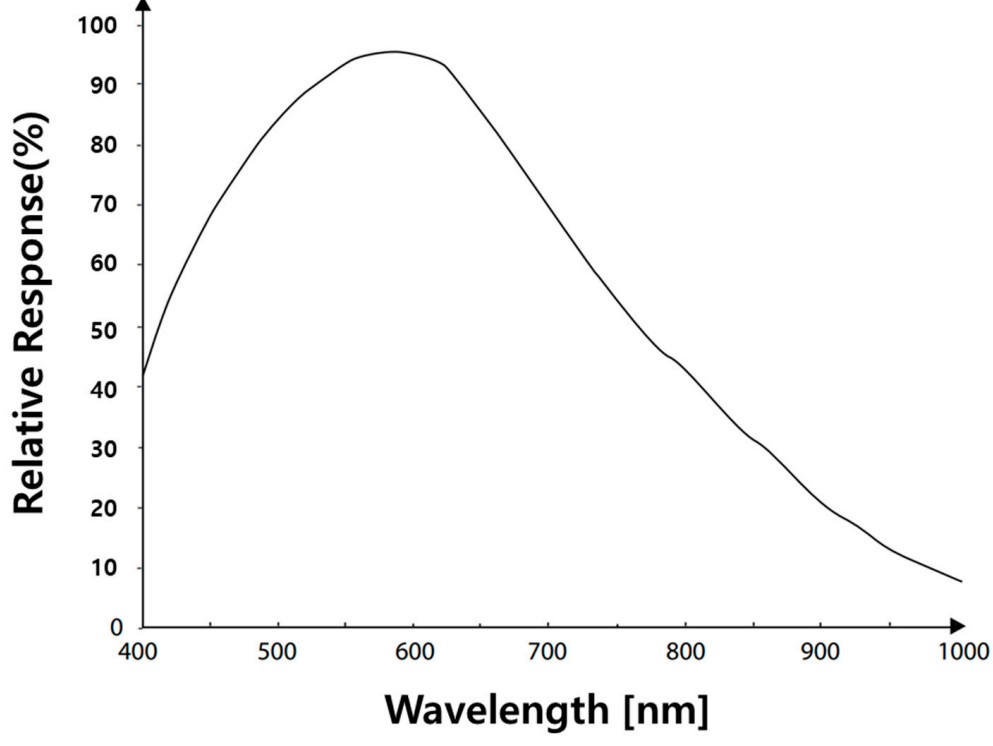

**Figure 1.** Spectral sensitivity of the imaging sensor, Sony ICX445.

The standard size of the passage width of the sweet pepper greenhouse was 1.6 m. The width between crops not interrupted by sweet pepper crops when moving from one passage to another was 1.0 m [29]. The planting density ranged from 3.3 to 3.5 stems/m$^2$ [30]. The

spacing of sweet pepper nutrient solution beds based on the planting density was within 0.3 m. An 8 mm lens was selected to acquire images of the sweet peppers planted at intervals of 0.3 (the minimum) to 1.0 m (the maximum). The size of the Sony ICX445 image sensor on the selected camera was 1/3". The field of view (FOV) according to the working distance using an 8 mm lens is shown in Figure 2. The distance between the pupils of human eyes ranges from about 60 to 75 mm [31]. Thus, the distance between cameras was set to 70 mm to acquire sweet pepper images (Figure 3). According to Figure 2, the area of the imaging system in this study overlapped from a maximum of 530 to a minimum of 230 mm. The average length of sweet pepper leaves was 223 mm. The average leaf width was 121 mm [32] and the average diameter of sweet pepper fruit was 80.92 mm, suitable for obtaining overlapping images of each sweet pepper part at the corresponding distance.

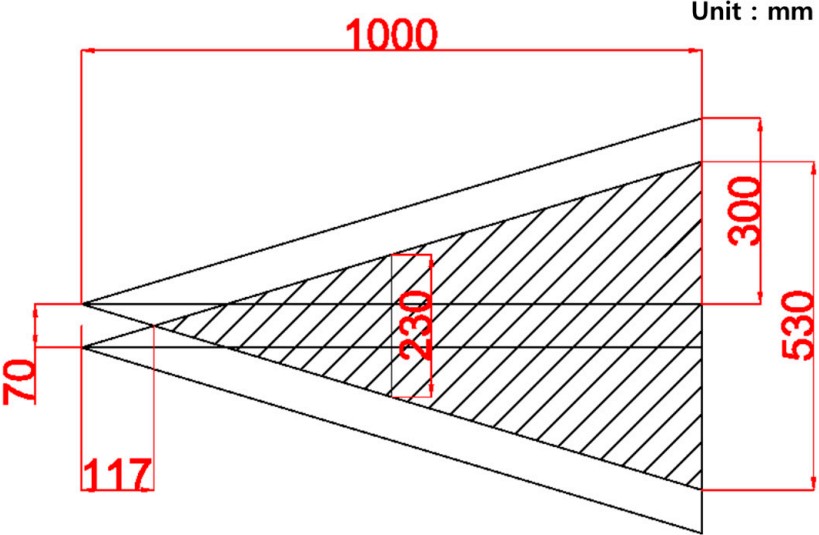

**Figure 2.** FOV of 8 mm lens and overlapping area of vision system.

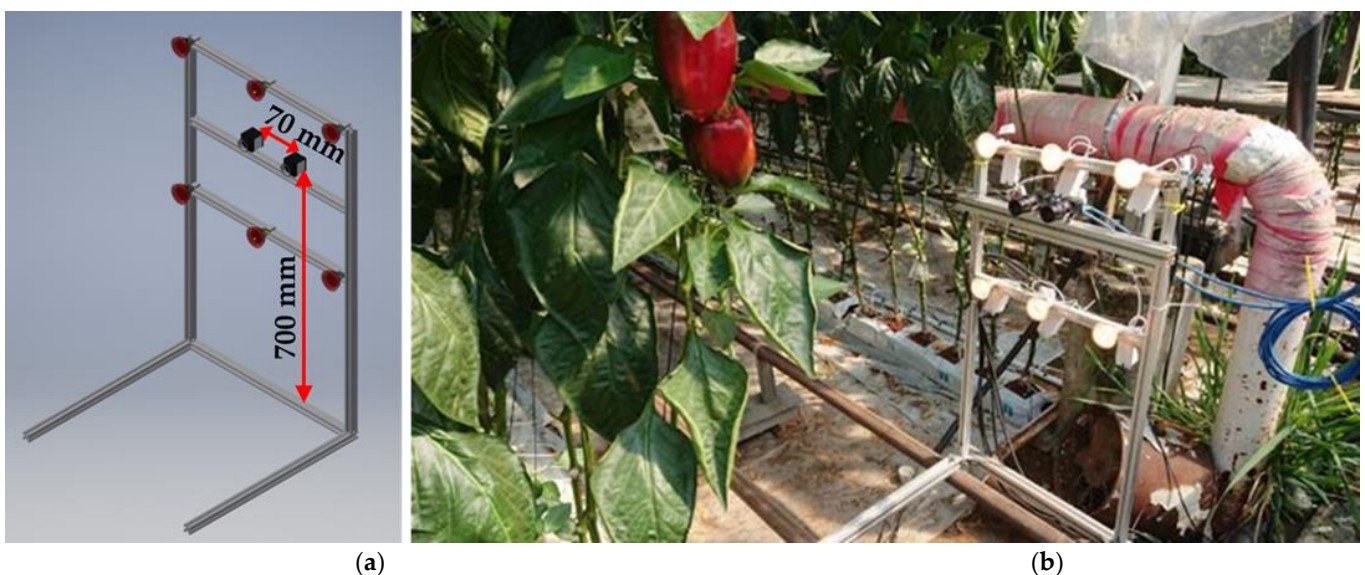

(**a**)    (**b**)

**Figure 3.** A structure for image acquisition (**a**) and image acquisition in the greenhouse (**b**).

The device for obtaining images of the sweet peppers was designed as shown in Figure 3a. Figure 3b shows image acquisition. The sweet pepper was planted in nutrient solution beds. The minimum height of the nutrient solution beds was 0.1 m and the camera's FOV at 1 m was 0.6 m. Thus, the height of the camera from the ground was set at 0.7 m. In addition, halogen lamps were selected as image light sources. Three lamps each

were installed on the upper side and the lower side for lighting. A halogen lamp 44870 WFL (Osram, Munich, Bavaria, Germany) was selected as an image light source to assist image acquisition in order to obtain an image from the 435–950 nm region. The selected lamp had a brightness of 680 lm of 50 W and a color temperature of 3000 K. Halogen lamps have a broad spectrum from visible light to infrared regions [33]. For image analysis using the NDVI, a band-pass filter from the 435 to 950 nm region of BP-Series (Midwest Optical Systems, Palatine, Illinois, USA) was used. Specifications of the band-pass filter used are shown in Table 2.

**Table 2.** Specifications of the band-pass filter used by the system.

| Model | Useful Range | FWHM [1] | Tolerance | Peak Transmission |
|-------|-------------|----------|-----------|-------------------|
| BP470 | 435−495 nm | 85 nm | +/−10 nm | >90% |
| BP500 | 440−555 nm | 248 nm | +/−10 nm | >85% |
| BP505 | 485−550 nm | 90 nm | +/−10 nm | >90% |
| BP525 | 500−555 nm | 80 nm | +/−10 nm | >90% |
| BP635 | 610−650 nm | 65 nm | +/−10 nm | >90% |
| BP660 | 640−680 nm | 65 nm | +/−10 nm | >90% |
| BP695 | 680−720 nm | 65 nm | +/−10 nm | >90% |
| BP735 | 715−780 nm | 90 nm | +/−10 nm | >90% |
| BP800 | 745−950 nm | 315 nm | +/−10 nm | >90% |

[1] FWHM: Full-Width Half-Maximum.

### 2.2. Sweet Pepper Part Classification Process

When growing sweet pepper, to promote its growth, reduce damage caused by pests and diseases, and increase fruit yield, it is necessary to remove old leaves at the lower part and suckers that occur between central stems and nodes [28]. In order to automate agricultural tasks such as pruning and harvesting, it is important to detect the branching point of the central stem and nodes, the sucker that is a shoot of new growth from the nook where a branch splits in two and a piece of the plant that gardeners remove in the pruning process. In this study, local features of sweet pepper were used for the classification of plant parts such as fruits, stems, leaves, and nodes from obtained images to differentiate them by each part. First, the harvesting image was pre-processed through the NDVI to remove noise such as the background image. Fruit, stem, leaf, and node were segmented through k-means clustering. Skeletonization of each segmented part was performed through a morphological method. Local features from the skeletonized image were then figured out using SIFT and SURF algorithms. Numbers of local features of fruits, stems, leaves, and nodes were then compared to confirm local features. The extracted local features were classified through the SVM and BP classifier algorithms. Their classification performances were compared. In addition, a CNN (Convolutional Neural Network), a type of deep neural network, was implemented to differentiate fruits, nodes, leaves, and suckers of sweet pepper. Its classification performance was compared to that of the SVM or BP classifier algorithm. Figure 4 shows the whole classification process for plant parts.

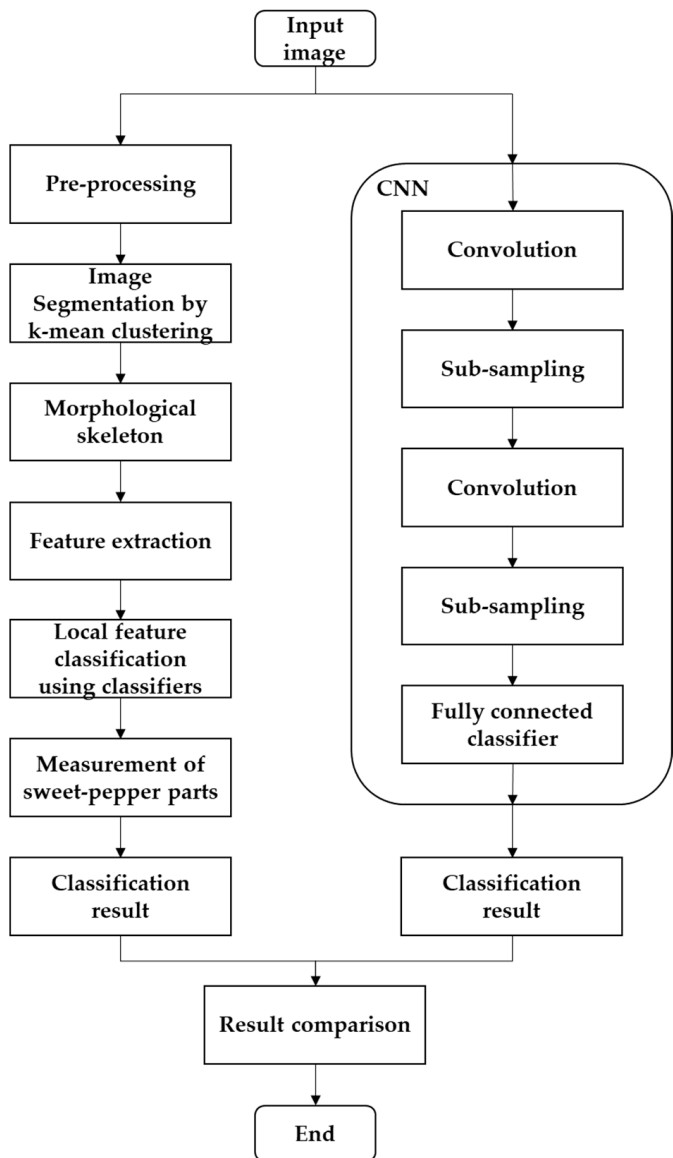

**Figure 4.** The classification process of sweet pepper plant parts.

2.2.1. Pre-Processing for Differentiation of Stems and Leaves from Backgrounds Using the NDVI

Silva et al. [14], Story et al. [34], and Yu [35] analyzed plants using their differences in light transmittance and reflectance according to plant wavelength. Figure 5 shows an image in the visible spectrum band and images using band-pass filters of 695, 735, and 800 nm.

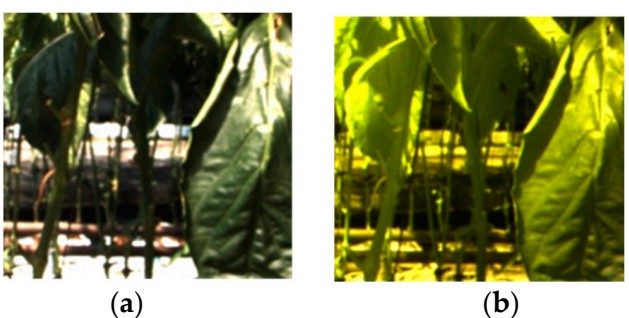

**(a)**　　　　　　　　　　　　　　　**(b)**

**Figure 5.** *Cont*.

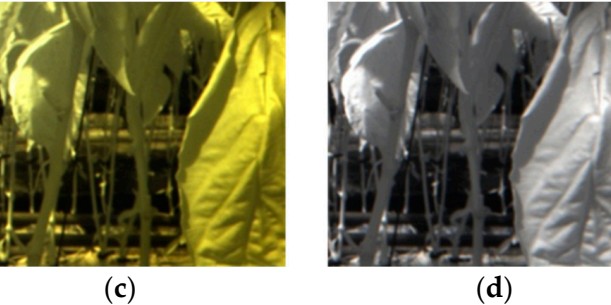

**Figure 5.** Spectrum images using band-pass filters of (**a**) visible spectrum, (**b**) 695, (**c**) 735, and (**d**) 800 nm.

Plants have a different reflectance for each wavelength band. Sweet pepper stems and leaves were separated from the background by analyzing images from 470 to 800 nm. The NDVI used the difference in reflectance for each wavelength band to distinguish plants from non-plants. As shown in Figure 6, 695, 735, and 800 nm sectors (hereinafter, group A) and 470, 500, 505, 525, 635, and 660 nm sectors (hereinafter, group B) were analyzed to select a wavelength band suitable for the imaging system of this study. Using Formula (1) based on the NDVI, differences between A and B sectors were investigated.

$$f = \frac{A - B}{A + B} \times 255 \tag{1}$$

where $A$ = image at 695, 735, and 800 nm; $B$ = image at 470, 500, 505, 525, 635, and 660 nm, and $f$ = NDVI image (Rouse et al. [36]).

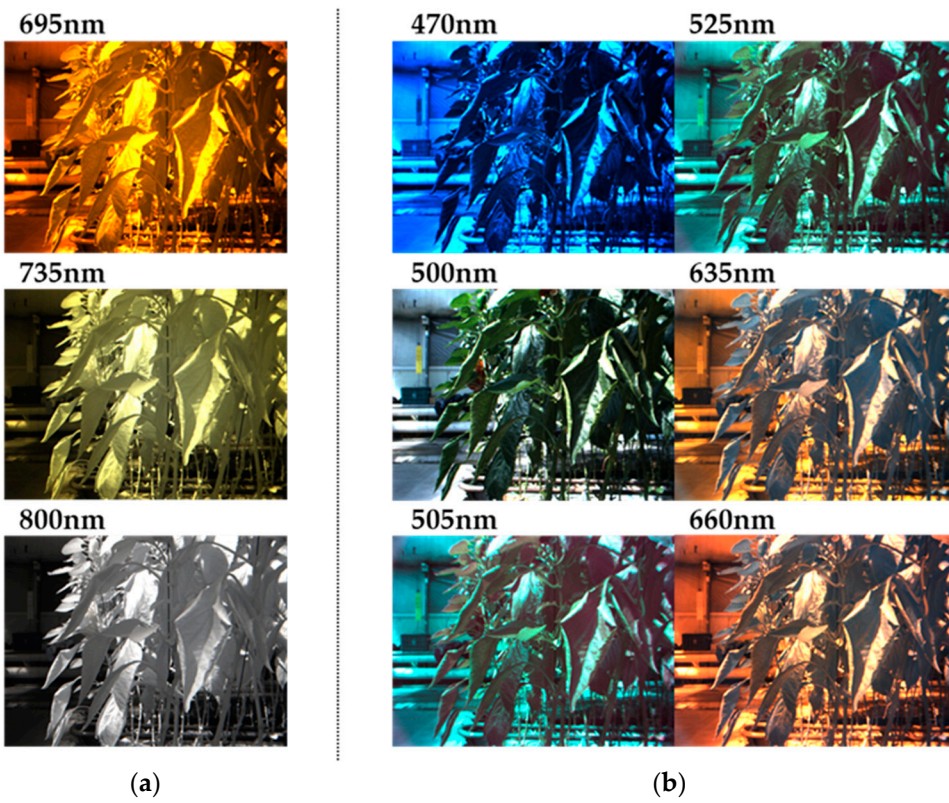

**Figure 6.** Image comparison between (**a**) images of 695, 735, and 800 nm wavelength and (**b**) images at 470, 500, 505, 525, 635, and 660 nm wavelength.

After obtaining the ratio of the difference and the sum of images of groups A and B, it was multiplied by 255, normalized to a size between 0 and 255, and then visualized. The quality of the normalized image was evaluated using Peak Signal-to-Noise Ratio (PSNR),

the maximum signal-to-noise ratio. Figure 7a is an image in the visible ray region, which is a reference for comparing image quality and differences of images by wavelength bands. Figure 7b shows the loss degree by PSNR.

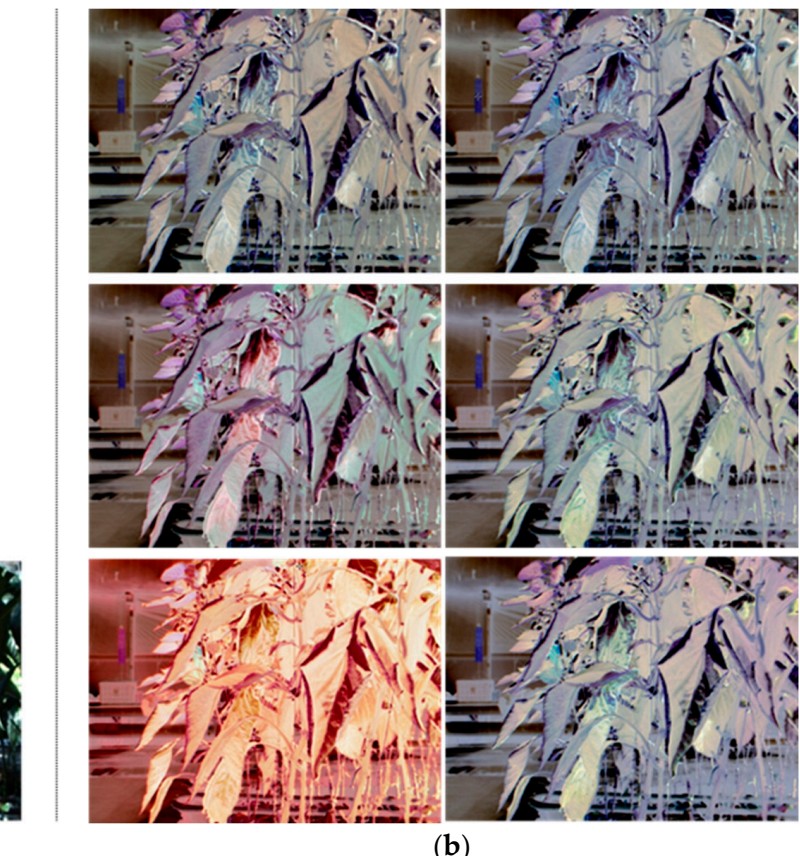

(**a**)  (**b**)

**Figure 7.** Reference image for PSNR comparison (**a**) and PSNR images by difference of wavelength (**b**).

PSNR was measured using a total of 180 NDVI images (10 images from each band). A value with a high average of PSNR from each band difference was selected as an image for the NDVI. An image with a higher PSNR value meant less distortion during image conversion to image of higher quality. Leaves and stems were separated in the sweet pepper harvesting image using differences between groups A and B with high PSNR values. Local features of leaves, stems, and nodes were then extracted from separated regions to detect sweet pepper leaves, stems, and nodes within the image.

### 2.2.2. Image Segmentation by K-Means Clustering

In this paper, the image that underwent the pre-processing of separating targeted objects from the background was segmented through k-means clustering, which was first proposed by Lloyd [37]. In the farm image, colors of the same objects can have various color distributions depending on nutrition and growth conditions, lighting, sunlight, and shadows. Therefore, if the image was segmented using a simple binarization method, it was difficult to accurately classify targeted objects. When segmentation by k-means clustering was performed, color segmentation of targeted objects was possible rather than performing segmentation using a single threshold value.

$$min_{m_k} \sum_{k=1}^{k} \sum_{x \in c_k} (x - m_k)^2 \tag{2}$$

where $x$ = image pixels to be clustered; $m_k$ = the center of clusters; $k$ = the number of randomly assigned clusters at the beginning.

In the clustering algorithm (Equation (2)), k-means clustering made a starting point at an arbitrary position as many as the initial number of k. Based on this, the distance between each data point was calculated using the Euclidean distance and clustered at the nearest initial point. A position at which the average point of the image pixels divided into clusters was obtained. It became a new center point. This process was repeated until the average position of all clusters did not change by performing clustering. In this paper, the number of clusters, k, was designated as 3 using HSV color information known to be stable upon lighting changes rather than RGB (Red, Green, Blue) color information. Three clusters were then obtained for fruits, stems, and leaves images with the background separated. For sweet pepper under cultivation, 400 sweet pepper cultivation images of 1288 × 964 pixels were obtained at 680–800 mm. The ratio of width and length was equally divided into 4 parts as shown in Figure 8. Numbers of fruits, stems, leaves, and suckers lost during image segmentation in each area were then measured.

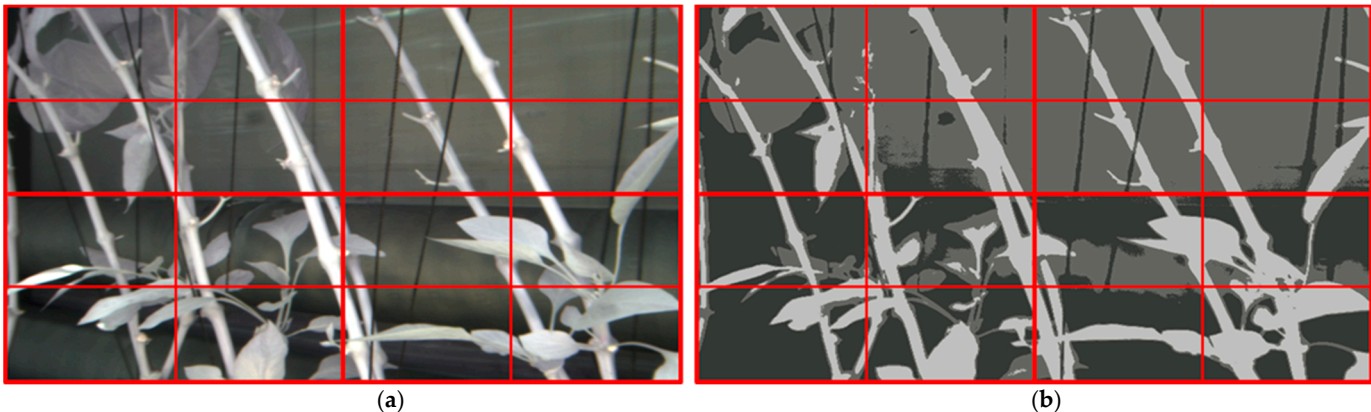

(**a**)           (**b**)

**Figure 8.** Pictures of each part before image segmentation (**a**) and after image segmentation (**b**).

### 2.2.3. Extracting Local Features

Chatbri et al. [38] compared local features of objects generated by contour detection and skeletonization of the image to extract the morphological features of objects in the binary image segmented by color. In this paper, SIFT [39] proposed by David G. Lowe and SURF [40] proposed by H. Bay were used to extract local features from 40 skeletonized images through a morphological method. By extracting and comparing the number of local features of each method, local features suitable for differentiating sweet pepper parts were selected. In this study, the curvature threshold of SIFT was set to be 10 and the control threshold was set to be 0.04. SURF constructed a window using Haar wavelets in the surrounding area based on the direction and intensity of key points found in the scale space. This window had local features that were strong against rotation and scale changes. Once SIFT local features were used, a 128-dimensional vector was obtained as a descriptor representing the point using an image gradient of 4 × 4 size in 8 directions. Using local features of the SURF, the gradients in the x-direction and y-direction were classified into 8 categories and accumulated to obtain a 128-dimensional vector as a descriptor representing key points. To use as learning factors for SVM and BP algorithms, coordinates, size, angle, response, octave, and class identification number found using SIFT and SURF were stored as key point information. The descriptor was stored as 128 real numbers. Among them, 128 descriptors of each key point were used as learning factors for SVM and BP algorithms. Figure 9 shows a sweet pepper cultivation image segmented by k-means clustering with local features extracted using skeletonization, SIFT, and SURF.

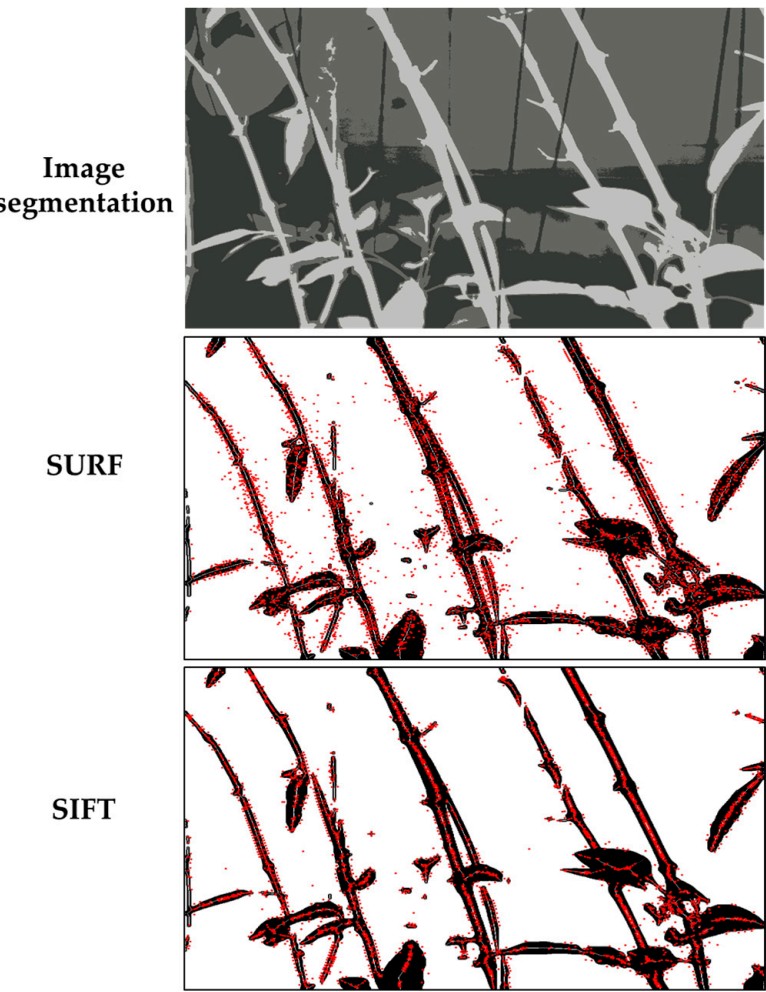

**Figure 9.** The image of local feature extraction processes using skeletonization, SIFT, and SURF.

### 2.2.4. Local Features Classification Using the SVM Algorithm and BP Algorithm

The method of classifying local features was determined by training and recognizing local features obtained in Section 2.2.3 using the SVM (Support Vector Machine) algorithm proposed by Vapnik [41] and the BP (Back Propagation) algorithm proposed by P. Werbos [42].

In this paper, the RBF (Radial Basis Function) kernel was used as the kernel function of SVM. The RBF (Radial Basis Function) kernel is expressed as Equation (3):

$$K(x,y) = exp\left(-\frac{\|x - y^2\|}{2\sigma^2}\right)$$ (3)

where $\sigma$ = Gaussian window width; $x$ = input vector; $y$ = input pattern, $K(x,y)$ = RBF kernel.

SVM is basically a binary classifier that solves two kinds of problems. Methods for expanding the M-class SVM include a one-to-many classification method and a one-to-one method. In this paper, M binary classifiers were performed using the one-to-many classification method. Here, the $i$-th binary classifier classified the class and the remaining M-1. In this way, classification for M times was performed and the highest value was classified.

$$k = \underset{i}{argmax}\ d_i(x)$$ (4)

where $d_i(x)$ = the decision hyperplane of the $i$-th binary classifier.

The *k* in Equation (4) indicates that the class with the highest value is classified from the M-class SVM classifiers. To classify local features of sweet pepper using the SVM algorithm, a total of 400 images were used (100 images each were used to determine local features of fruits, stems, leaves, and suckers). For each class, 60 were used for training and 40 were used for validation.

Figure 10 shows the structure of the BP neural network used in this paper. The connection strength w was initially set to a value ranging from −0.5 to maximum 0.5 and v was initially set to a random value ranging from minimum −0.5 to +0.5. v was initialized by Equation (5) at the start of learning:

$$v^{new} = \frac{\beta v^{old}}{\|v^{old}\|}, \quad \beta = 0.7^n \sqrt{p} \tag{5}$$

where *n* = the number of nodes at input layer; *p* = the number of nodes at hidden layer.

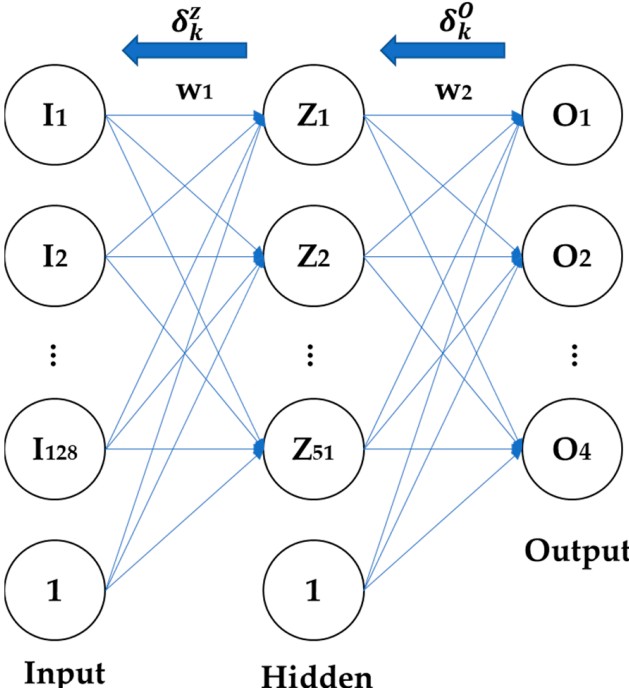

**Figure 10.** The structure of the backpropagation algorithm.

The number of nodes of the input layer was 129, including bias nodes and 128 descriptors of local features calculated by the gradient histogram of local features. If the input pattern space is *n*-dimensional and the number of nodes in the hidden layer is *p*, the maximum number M of linearly separable regions can be obtained through Equation (6) when *k* is smaller than *p*. If n is greater than or equal to *p*, it can be obtained through Equation (7). Therefore, the number p of nodes in the hidden layer for solving the problem of M linearly separable regions can be obtained using Equation (8):

$$(case\ 1)\ if\ k < p,\quad M = \sum_{k=0}^{n} {}_pC_k, \tag{6}$$

$$(case\ 2)\ if\ n \geq p,\quad M = 2^p, \tag{7}$$

$$p = \log_2 M \tag{8}$$

where *n* = the number of nodes at input layer; *p* = the number of nodes at hidden layer; *M* = the maximum number of regions capable of linear separation.

The number of nodes (*p*) in the hidden layer was set at 51, including bias nodes. The number of nodes in the output layer was set at 4 because there should be one node for each class of fruits, stems, leaves, and suckers. The activation function used a bipolar sigmoid function with a value between −1 and 1. The learning rate was set to be 0.001. The

maximum error was set to be 0.1. Learning was terminated when an error of less than 0.1 was obtained. In case the error did not become smaller than 0.1, learning was terminated when the number of learning generations (epoch) exceeded 500. A total of 400 images were used to classify local features of sweet peppers using the BP algorithm. Images were classified using the descriptor of key points for images of fruits, nodes, leaves, and suckers as learning factors of the SVM and the BP algorithm. Classification results were evaluated using precision, recall, F-measure, and accuracy.

2.2.5. Partial Classification Performance Experiment Using Deep Neural Network

In this paper, a convolutional neural network (CNN), a kind of deep neural network, was implemented to differentiate fruits, stems, leaves, and suckers of plants and classify each part of the sweet pepper. The CNN in this paper had 16 hidden layers using the focal loss method of Lin et al. [43]. The bias was 0 and the weight was a random Gaussian weight except for the final layer. The bias of the last layer was calculated using Equation (9):

$$bias = -\log\left(\frac{1-\pi}{\pi}\right) \tag{9}$$

where $\pi$ = the confidence value for the foreground of labels of all anchors in the initial learning (0.01).

In Equation (10), the focal loss function was calculated with $\gamma = 2$ as the rectified linear unit (ReLU) activation function, and the learning rate was set to be 0.01.

$$FL(p_t) = -(1-p_t)^\gamma \log(p_t), \quad \gamma \geq 0$$

$$p_t = \begin{cases} p & if\ y = 1 \\ 1-p & otherwise \end{cases} \tag{10}$$

where $y$ = the ground-truth class (1 or $-1$); $p$ = the class probability predicted by the model (if $y = 1$); $\gamma$ = the focal loss function.

In order to learn each part of the sweet pepper, as shown in Figure 11, a total of four types of sweet pepper were classified into fruits, stems, leaves, and suckers. A total of 300 brightness images were used (75 for each class mentioned above). A total of 100 images of sweet pepper cultivation were used to verify the performance of the deep neural network that was learned. Classification results were evaluated using precision, recall, F-measure, and accuracy. In order to confirm the classification performance for suckers, stems, leaf, and fruits of sweet peppers, a good method for differentiating plant parts between the SVM and BP algorithms was selected. The performance of the better one of the two ways (SVM and BP algorithms) was then compared to that of the deep neural network.

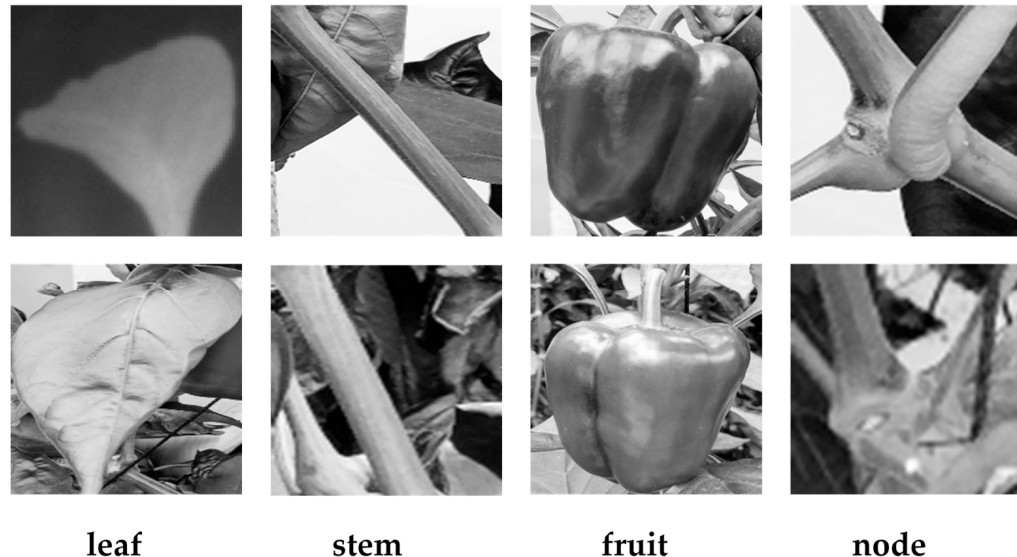

**Figure 11.** Example of learning data for leaf, sucker, fruit, and node.

## 3. Results and Discussion

### 3.1. Part Classification Results

#### 3.1.1. Results of Pre-Process for Differentiation of Stems, Leaves, and Backgrounds Using NDVI

Figure 12 shows the transformation of the difference between images in 695, 735, and 800 nm bands and the images in the 470, 500, 505, 525, 635, and 660 nm bands through Equation (1). The quality of the transformed image was evaluated through the PSNR (Peak Signal-to-Noise Ratio) value. The higher the PSNR value, the lower the loss. Table 3 shows the average of PSNR values of images in the 695, 735, and 800 nm bands and the difference images of 10 images in the 470, 500, 505, 525, 635, and 660 nm bands. Among the average PSNR values measured in the 10 images with a total of 18 band differences, the image obtained by transforming the difference between 735 and 660 nm showed the highest value of 871,625. The higher the PSNR value, the lower the loss compared to the original image. Therefore, the difference between 735 and 660 nm confirmed image with the least image loss compared to the visible light region. Stems, leaves, and the background were distinguished using the transformed image difference between 735 and 660 nm.

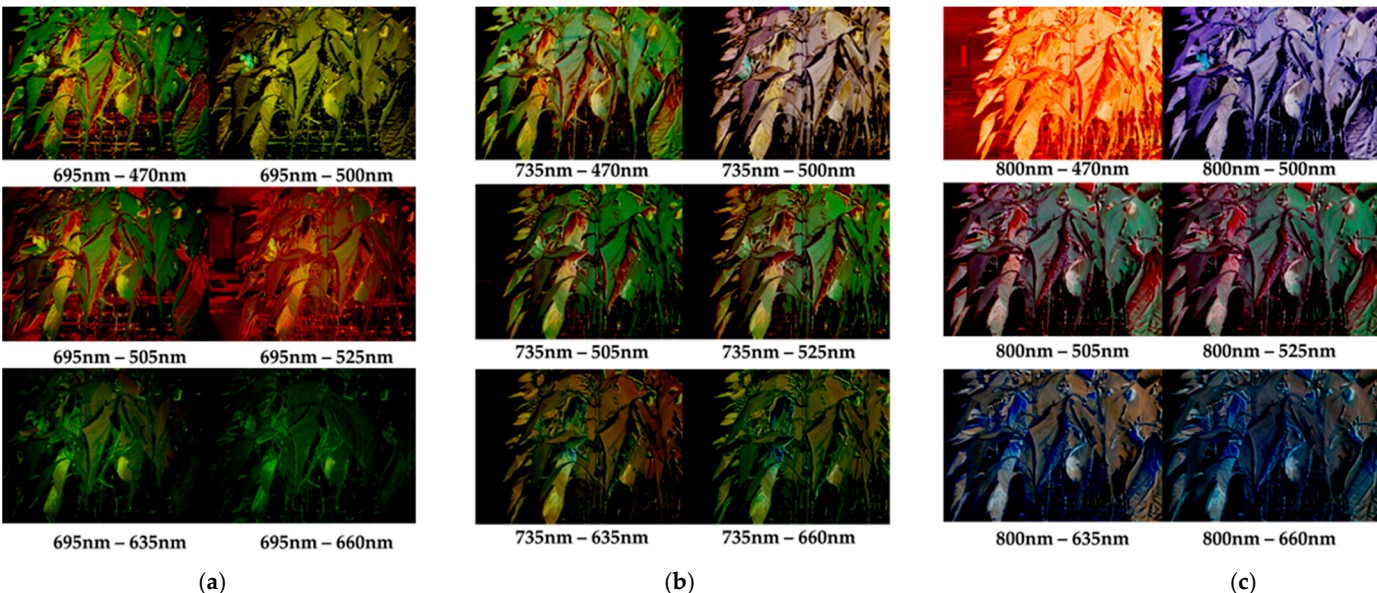

**Figure 12.** Differences of visible spectrum image at 695 (**a**), 735 (**b**), and 800 nm (**c**) wavelength by 470, 500, 525, 635, 660 nm.

**Table 3.** PSNR values at different wavelengths.

| Wavelength | 695 nm | | 735 nm | | 800 nm | |
|---|---|---|---|---|---|---|
| | Mean | SD [1] | Mean | SD | Mean | SD |
| 470 nm | 8.33309 | 0.0229 | 5.46441 | 0.0949 | 8.62191 | 0.0339 |
| 500 nm | 8.20943 | 0.0294 | 6.51014 | 0.0837 | 7.81063 | 0.0306 |
| 505 nm | 8.30828 | 0.0118 | 7.75226 | 0.0999 | 8.68881 | 0.0673 |
| 525 nm | 8.2368 | 0.0971 | 7.57739 | 0.0283 | 7.53663 | 0.0667 |
| 635 nm | 7.98845 | 0.0248 | 7.62452 | 0.0393 | 7.66518 | 0.0538 |
| 660 nm | 7.84794 | 0.0298 | 8.71625 | 0.0315 | 7.70154 | 0.0178 |

[1] SD: standard deviations.

#### 3.1.2. Image Segmentation Results by K-Means Clustering

Plenty (a red sweet pepper variety), President (a yellow sweet pepper variety), and Derby (an orange sweet pepper variety) varieties were used in this study. In the average image of the fruit area, the fruit and the background were differentiated by back-projecting the histogram of color and saturation. For stems and leaves, the canopy of sweet pepper and the background were differentiated using the NDVI for the images of 735 and 660 nm bands.

The result of image segmentation using k-means clustering with the number 3 clusters separated from the background is shown in Figure 13. To determine losses of fruits, stems, leaves, and suckers in the image through image segmentation, an image of a sweet pepper at 1288 × 964 pixels was divided into four parts horizontally and vertically. Numbers of fruits, stems, leaves, and suckers damaged in the area were measured.

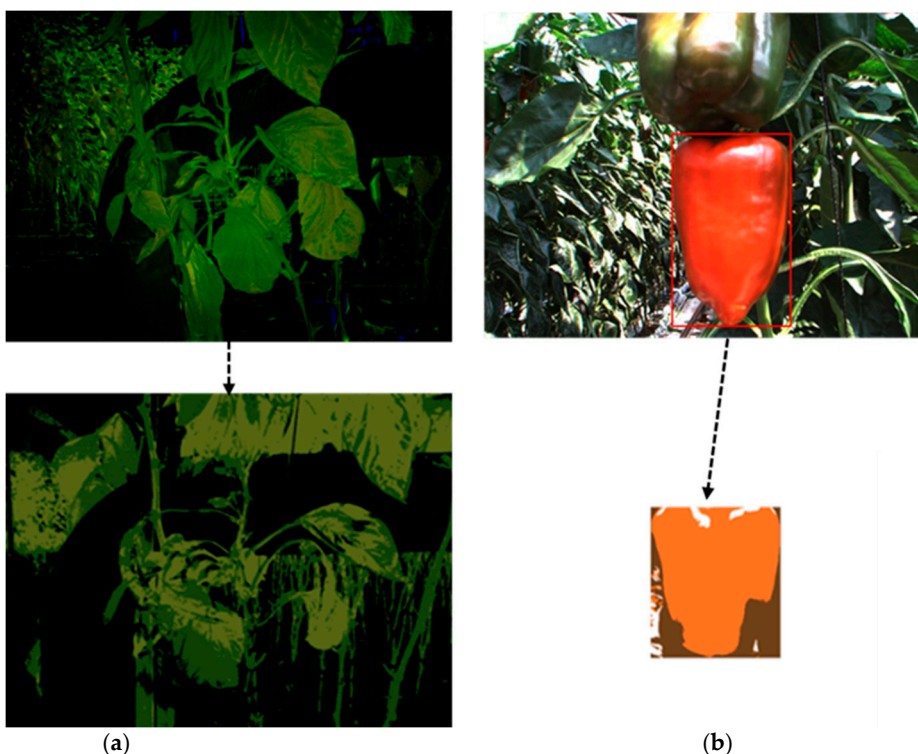

(**a**) (**b**)

**Figure 13.** Image segmentation of steam, leaves, suckers (**a**), and fruits (**b**) by k-means clustering.

Table 4 shows actual numbers of fruits, stems, leaves, and suckers in the cultivation image and numbers of fruits, stems, leaves, and suckers obtained through the image segmentation process. As a result, in the case of fruits, through the image segmentation process, the images of 6 fruits, 1421 stems, 2188 leaves, and 37 suckers were removed from the background.

**Table 4.** Numbers of parts retained after image segmentation.

| Parts | Actual Number | Segmented Number |
|---|---|---|
| Fruit | 682 | 676 |
| Sucker | 345 | 308 |
| Leaf | 8169 | 5981 |
| Node | 11,856 | 10,435 |

### 3.1.3. Results of Local Feature Extraction

Using k-means clustering, the targeted region and the background were image segmented. The image was modified so that local features could be obtained through morphological skeletonization. Numbers of local features of SIFT and SURF were compared in images of 80 × 80 pixels in width and length of 30 fruits, nodes, leaves, and suction regions, respectively. Figure 14a shows numbers of local features extracted using SIFT. For fruits, the median number and average number of local features were 121 and 123, respectively, with a maximum value of 129 and a minimum value of 111. In the case of suckers, the median number and average number of local features were 105 and 101, respectively, with a maximum value of 123 and a minimum value of 81. In the case of leaves, the median

number and average number of local features were 103.5 and 104, respectively, with a maximum value of 115 and a minimum value of 92. For stems, the median, the average, the maximum, and the minimum were 111.5, 113, 133, and 99, respectively.

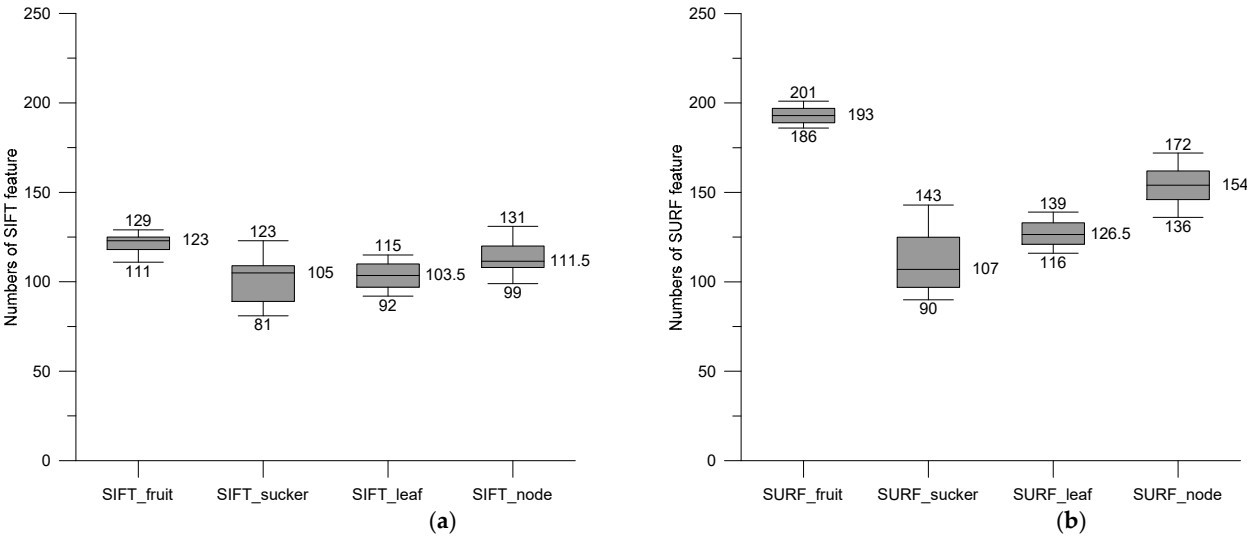

**Figure 14.** Numbers of SIFT features (**a**) and SURF features (**b**).

Figure 14b shows the number of local features extracted using SURF. For fruits, the median number and the average number of local features were the same at 193, with a maximum value of 201 and a minimum value of 186. In the case of suckers, the median number and average number of local features were 107 and 112, respectively, with a maximum value of 143 and a minimum value of 90. In the case of leaves, the median number and average number of local features were 126.5 and 127, respectively, with a maximum value of 139 and a minimum value of 116. For stems, the median, the average, the maximum, and the minimum were 154, 153, 172, and 136, respectively. In the case of suckers, numbers of local features of SIFT and SURF were extracted from overlapping ranges. The average, maximum, and minimum values showed high SURF. In the case of fruits, leaves, and stems, the number of local features of SURF was higher than that of SIFT without overlapping ranges. Therefore, the SURF algorithm was selected as the classification method for extracting local features of sweet pepper fruits, stems, leaves, and suckers.

### 3.1.4. Classification Results of Local Features Using SVM and BP Algorithms

Fruit and background were differentiated by back-projection of the histogram of color and saturation of the average image of the fruit area. In the case of stems and leaves, the canopy and the background of sweet pepper were differentiated using the NDVI operation of images at 735 and 660 nm band wavelengths. After segmenting the image, local features of fruits, stems, leaves, and suckers were extracted using SURF from skeletonized images and differentiated using the SVM algorithm and the BP algorithm. Performances of these two algorithms were then compared.

A total of 100 images of each region were used for features of the SURF region of fruits, stems, leaves, and suckers. As a result of classifying a total of 400 images through the SVM algorithm using the RBF kernel, the accuracy was 63.75%. Table 5 shows results of classification using SVM for fruits, stems, leaves, and suckers. The precision for fruit was 75% and the recall rate was 73.5%. The precision for stems was 68% and the recall rate was 61.26%. The precision for leaves was 60% and the recall rate was 57.69%. The precision for suckers was 52% and the recall rate was 62.65%. F-measure results for fruits, stems, leaves, and suckers were 74.26, 64.46, 58.82, and 56.83%, respectively.

**Table 5.** Results of SVM classifier using RBF kernel.

| Parts | | True Condition | | | |
| --- | --- | --- | --- | --- | --- |
| | | Fruit | Node | Leaf | Sucker |
| Predicted condition | Fruit | 75 | 0 | 25 | 0 |
| | Node | 0 | 68 | 11 | 21 |
| | Leaf | 27 | 3 | 60 | 10 |
| | Sucker | 0 | 40 | 8 | 52 |

Table 6 shows results of 178,750 features collected from 400 images of each class of fruit, stem, leaf, and sucker into a training set and a validation set used to train the BP algorithm. SURF region features were extracted from 400 images including fruits, stems, leaves, and suckers. Of a total of 178,750 features, 10% were used for training. As a result of repeating the learning for 500 generations (epoch) with the learning rate set to be 0.001, the final learning result was 98.46%.

**Table 6.** Number of features used for learning.

| Parts | Training Feature | Test Feature |
| --- | --- | --- |
| Fruit | 694 | 6250 |
| Node | 10,720 | 96,478 |
| Leaf | 6144 | 55,298 |
| Sucker | 317 | 2849 |
| Total feature | 17,875 | 160,875 |

The accuracy was 95.95% as a result of classifying through the BP algorithm using a total of 400 images with 100 images for each class targeting SURF local features of fruits, stems, leaves, and suckers. Table 7 shows results of classification using the BP algorithm for local features of fruits, stems, leaves, and suckers. The accuracy for local features of fruits was 95.27% and the recall rate was 96.18%. The accuracy for local features of stems was 96.26% and the recall rate was 97.60%. The accuracy for local features of leaves was 95.57% and the recall rate was 93.42%. The accuracy for local features of suckers was 94.47% and the recall rate was 90.94%. F-measure results for local features of fruits, stems, leaves, and suckers were 95.72, 96.93, 94.48, and 92.67%, respectively.

**Table 7.** Results of the BP algorithm.

| Parts | | True Condition | | | |
| --- | --- | --- | --- | --- | --- |
| | | Fruit | Node | Leaf | Sucker |
| Predicted condition | Fruit | 6616 | 1 | 325 | 2 |
| | Node | 102 | 103,196 | 3758 | 142 |
| | Leaf | 154 | 2416 | 58,718 | 154 |
| | Sucker | 7 | 115 | 53 | 2991 |

The accuracy of the BP algorithm was 95.95%, which was higher than that of the SVM at 63.75%. Accuracy, recall, and F-measure were all higher with the BP algorithm than those with the SVM. When the BP algorithm was used to classify SURF local features of fruits, stems, leaves, and suckers, it showed a higher performance than the SVM.

### 3.1.5. Results of Local Feature Performance Experiment Using the Deep Neural Network

When the deep neural network was trained repeatedly for five generations using the focal loss method, the accuracy reached 99.9%. As a result of learning through the learning system described in this study, it took an average of 2000 s to repeat the first generation. Of a total of 400 images, 75% were used for training.

As a result of classifying through a deep neural network using 400 images (100 images for each class of fruit, stem, leaf, and sucker), the accuracy was 92.93%. Table 8 shows results of classification using a deep neural network for fruits, stems, leaves, and suckers.

The accuracy for fruits was 99.5% and the recall rate was 97.79%. The accuracy for stems was 87.75% and the recall rate was 87.31%. The accuracy for leaves was 90.50% and the recall rate was 99.45%. The accuracy for suckers was 87.25% and the recall rate was 87.69%. F-measure results for fruits, stems, leaves, and suckers were 98.63, 87.53, 94.76, and 87.47%, respectively. In the process of acquiring the 3D shape as a two-dimensional image, the shape of the sucker and the node was not clearly distinguished, so 49 out of 400 nodes were classified as suckers, and 51 out of 400 suckers were classified as nodes. In the case of leaves, 29 leaves in images were not recognized as leaves. The more samples used for training with the CNN algorithm, the better the performance. Therefore, in order to solve the problem of accuracy deterioration due to confusion between nodes and sucker and the problem of not recognizing leaves that occurred in this paper, it is necessary to acquire images of nodes, suckers, and leaves from various directions and angles and increase the number of samples used for training.

**Table 8.** Result of the CNN algorithm.

| Parts | | True Condition | | | |
|---|---|---|---|---|---|
| | | Fruit | Node | Leaf | Sucker |
| Predicted condition | Fruit | 398 | 0 | 2 | 0 |
| | Node | 0 | 351 | 0 | 49 |
| | Leaf | 9 | 0 | 362 | 0 |
| | Sucker | 0 | 51 | 0 | 349 |

*3.2. Results of Comparing Classification Performances between BP and CNN for Sweet Pepper Parts*

Table 9 shows results of comparing classification performances for sweet pepper parts. The classification performance of the BP algorithm was the result of summing proportions of each region lost in the image segmentation process. In the case of plant part classification using the BP algorithm, the recognition success rate was 94.44% for fruits, 84.73% for stems, 69.97% for leaves, and 84.34% for suckers. In the case of plant part classification using the deep neural network, the recognition success rate was 99.50% for fruits, 87.75% for stems, 90.50% for leaves, and 87.25% for suckers. In the case of plant part classification using the BP algorithm, the separation of fruit and background and separation of canopy and background were performed before fruit classification. In this process, 99.12% of fruits, 88.01% of stems, 73.22% of leaves, and 89.28% of suckers were preserved during image segmentation. In the image segmentation process, some regions were removed together with the background. For this reason, the performance of plant part classification using the BP algorithm might be lower than the performance of classifying local features of each part using the BP algorithm. Therefore, when the classification performance of the BP algorithm was compared to that of the deep neural network, the performance of the deep neural network was judged to be superior to that of the BP algorithm.

**Table 9.** Classification performance comparison between the BP algorithm and CNN.

| Performance | Fruit (%) | Node (%) | Leaf (%) | Sucker (%) |
|---|---|---|---|---|
| BP Algorithm | 94.44 | 84.73 | 69.97 | 84.34 |
| CNN | 99.5 | 87.75 | 90.50 | 87.25 |

## 4. Conclusions

As a pre-processing step for plant part classification, a method of separating stem and leaf regions from the background using the NDVI based on the difference between a 735 nm band image and a 660 nm band image was applied. In addition, the color of the region of interest separated from the background was classified using k-means clustering. Through this, it was possible to segment the image of the region of interest. To find local features in an image in which the interested region was segmented using k-means

clustering, morphological skeletonization was performed for the image to find the skeleton of the region of interest. As a result of extracting local features using SIFT and SURF, it was confirmed that local feature extraction using SURF could lead to more local features. Therefore, the SURF method was selected to detect local features of fruits, stems, leaves, and suckers in this study.

To classify SURF local features of fruits, stems, leaves, and suckers, local features were classified using SVM and BP algorithms with the radial basis function kernel. Their performances were then compared. Results confirmed that the accuracy of the BP algorithm was 95.96%, which was higher than that of the SVM at 63.75%.

Fruits, stems, leaves, and suckers were classified using the CNN (Convolutional Neural Network), a kind of deep neural network. Its performance was then compared to that of the BP algorithm. As a result of comparing the performances for classifying each part of sweet pepper, it was confirmed that the performance of the CNN was superior to that of the BP algorithm. If further research, such as stereo vision, to calculate distance and the mechanism of a robot-arm to perform agricultural works is carried out, the proposed method in this study could be applied to an image processing algorithm for the multipurpose cultivation robot system.

**Author Contributions:** Conceptualization, B.L., D.K., Y.C., D.-C.K. and D.-H.L.; Methodology, B.L. and D.-H.L.; Software, B.L.; Validation, B.L., D.K. and Y.C.; Formal analysis, B.L.; Investigation, B.L., D.K. and Y.C.; Resources, B.L., D.K. and Y.C.; Data curation, Y.C.; Writing—original draft preparation, B.L., D.K. and Y.C.; Writing—review and editing, B.L., Y.C. and D.-H.L.; Visualization, Y.C.; Supervision, Y.C.; Project administration, Y.C.; Funding acquisition, Y.C. All authors contributed almost equally to all aspects of this research. All authors have read and agreed to the published version of the manuscript.

**Funding:** This work was supported by Korea Institute of Planning and Evaluation for Technology in Food, Agriculture and Forestry (IPET) through the Agriculture, Food and Rural Affairs Convergence Technologies Program for Educating Creative Global Leader Program funded by the Ministry of Agriculture, Food and Rural Affairs (MAFRA) (320018-2). It was also supported by a grant (NRF-2021R1G1A1012778) of the National Research Foundation of Korea (NRF) funded by the Korean government (MSIT, Ministry of Science and ICT). This research was supported by the "Research Base Construction Fund Support Program" funded by Jeonbuk National University in 2020.

**Institutional Review Board Statement:** Not applicable.

**Informed Consent Statement:** Not applicable.

**Data Availability Statement:** Not applicable.

**Conflicts of Interest:** The authors have no conflict of interest to disclose relevant to this study.

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
