# Peer review of "Comparing Performances of CNN, BP, and SVM Algorithms for Differentiating Sweet Pepper Parts for Harvest Automation"

_applsci, doi:10.3390/app11209583_

Round 1

Reviewer 1 Report

Page 1, Introduction: The authors start the introduction with the following sentence: “In agricultural work such as weeding and pruning, the technology to distinguish and classify plant parts is necessary for automation of various agricultural tasks.” What are the advantages of automation? What does automation of agricultural tasks accomplish or why is automation of agricultural tasks desired? I would think automation alleviates the labor burden and thus the labor cost associated with weeding and pruning tasks in agriculture. There may be other good reasons for automation. The authors need to explain more the advantages of automation. This would make the findings of their manuscript more valuable.

Page 1, Lines 43 – 45. The authors indicate that machine recognition is a useful tool for plant recognition and identification but state in lines 43 – 45 that imaging of plant parts has been limited thus far to controlled experimental settings. What applications have made machine recognition useful in real world agricultural settings? Has this technology been shown to be practical in the real world, and if so what are some examples?

Page 2, Lines 82 – 83: The authors chose to develop an image processing algorithm to classify parts of sweet peppers. Why did the authors choose sweet peppers for their analysis? Why not some other fruit or vegetable? Was there a valid reason for choosing sweet peppers or was the choice arbitrary?

Page 5, Lines 156 – 157: The authors refer to the sucker as “… parasitic at the branching point and the leaf and stem nodes at the lower part of the sweet pepper.” It would be good if the authors had a better definition for what a “sucker” is. I as the reader had to look online for a valid definition of a sucker. I found it was a shoot of new growth out of the nook where a branch splits in two. It’s a piece of the plant that gardeners remove in the pruning process.

Page 18, Table 9: Classification accuracy percentages are reported in Table 9 and throughout the text for various algorithms. However, there is no point of reference given on how these classification performance percentages stack up with others in the literature or if they are acceptable or too low. In the classification performance comparison between BP and CNN, the percentages are in the 90s for fruit but are at or below 90 for the other plant part categories. Earlier in the manuscript on page 2, the authors note that 95% recognition is comparable to human recognition accuracy. Do the lower classification performance percentages for the other plant parts such as the node, leaf, and sucker pose a problem? Are these percentages acceptable relative to what can be accomplished with humans or do they need to be increased to make the imaging system concept practical in a real world situation? What would be the consequences of “false positives” or “false negatives”?

Conclusion Section, Page 18: The conclusion section is a brief summary of the findings reported in the “Results and Discussion” section. It would be good if the authors provided an explanation of what their results mean in the conclusions section. What do their results imply about the ability to accurately image plant parts? How does accurate imaging of plant parts benefit the ability to automate the pruning process for sweet peppers, and can this imaging be applied to other fruit and vegetable products?

Author Response

Dear Reviewers and Editors,

We truly appreciate the editor and reviewers for giving us an opportunity to revise our work and have kind comments to improve our manuscript.

Thank you for the good suggestions to improve our manuscript. Below we have copied the text of those suggestions and added our response in italics (with red). We have incorporated manuscript changes based on almost all of these comments. Where we have given line numbers below, they refer to the version of the MS Word document without tracked changes.

Regards,

Yongjin Cho and co-authors

----------------------------------------------------

Response to comments: applsci-1413078

Reviewer #1

Comments:

Point 1:  

Page 1, Introduction: The authors start the introduction with the following sentence: “In agricultural work such as weeding and pruning, the technology to distinguish and classify plant parts is necessary for automation of various agricultural tasks.” What are the advantages of automation? What does automation of agricultural tasks accomplish or why is automation of agricultural tasks desired? I would think automation alleviates the labor burden and thus the labor cost associated with weeding and pruning tasks in agriculture. There may be other good reasons for automation. The authors need to explain more the advantages of automation. This would make the findings of their manuscript more valuable.

Response 1: We added sentences (line 32-34) in the revised manuscript as you advised.

Point 2:  

Page 1, Lines 43 – 45. The authors indicate that machine recognition is a useful tool for plant recognition and identification but state in lines 43 – 45 that imaging of plant parts has been limited thus far to controlled experimental settings. What applications have made machine recognition useful in real world agricultural settings? Has this technology been shown to be practical in the real world, and if so what are some examples?

Response 2: We added sentences (line 46-52) in the revised manuscript as you advised.

Point 3:  

Page 2, Lines 82 – 83: The authors chose to develop an image processing algorithm to classify parts of sweet peppers. Why did the authors choose sweet peppers for their analysis? Why not some other fruit or vegetable? Was there a valid reason for choosing sweet peppers or was the choice arbitrary?

Response 3: We revised sentences (line 92-95) in the revised manuscript as you advised.

Point 4:

Page 5, Lines 156 – 157: The authors refer to the sucker as “… parasitic at the branching point and the leaf and stem nodes at the lower part of the sweet pepper.” It would be good if the authors had a better definition for what a “sucker” is. I as the reader had to look online for a valid definition of a sucker. I found it was a shoot of new growth out of the nook where a branch splits in two. It’s a piece of the plant that gardeners remove in the pruning process.

Response 4: We revised sentences (line 165-167) in the revised manuscript as you advised.

Point 5:

Page 18, Table 9: Classification accuracy percentages are reported in Table 9 and throughout the text for various algorithms. However, there is no point of reference given on how these classification performance percentages stack up with others in the literature or if they are acceptable or too low. In the classification performance comparison between BP and CNN, the percentages are in the 90s for fruit but are at or below 90 for the other plant part categories. Earlier in the manuscript on page 2, the authors note that 95% recognition is comparable to human recognition accuracy. Do the lower classification performance percentages for the other plant parts such as the node, leaf, and sucker pose a problem? Are these percentages acceptable relative to what can be accomplished with humans or do they need to be increased to make the imaging system concept practical in a real world situation? What would be the consequences of “false positives” or “false negatives”?

Response 6: We added sentences (line 494-502) in the revised manuscript as you advised.

Point 6:

Conclusion Section, Page 18: The conclusion section is a brief summary of the findings reported in the “Results and Discussion” section. It would be good if the authors provided an explanation of what their results mean in the conclusions section. What do their results imply about the ability to accurately image plant parts? How does accurate imaging of plant parts benefit the ability to automate the pruning process for sweet peppers, and can this imaging be applied to other fruit and vegetable products?

Response 6: We added sentences (line 548-551) in the revised manuscript as you advised.

Reviewer 2 Report

Congratulations!

Great work done with very difficult subject of vision quality control system. Parac is very implementation-oriented, patent-pending, for practical use. No methodological reservations, a well-prepared review of the literature, a correctly described methodology of the work, clearly presented results and conclusions.

Please expand the descriptions of the algorithm revision systems from the literature review from the points of biography 3-10. please specify their description. Appearing abbreviations like RGB, SURF, SIFT ... - should be explained immediately after the first appearance of the abbreviation.Figure 3a - Improve photo qualityWhat is the letter f in Formula 1? What is this Formula, what is its source?Same with Formula 2, 3 ...Explain all symbols in the formulas, e.g. 6, 7, 8 ...Generally, Forlumas are unexplained by the Authors.Figure 11. Why are the sucker and node not sharp?Consider whether in Figure 12 all photos are necessary. What do they contribute, how do they improve the quality of the manuscript?I advise you to think about the necessity of a native speaker of the manuscript

Author Response

Dear Reviewers and Editors,

We truly appreciate the editor and reviewers for giving us an opportunity to revise our work and have kind comments to improve our manuscript.

Thank you for the good suggestions to improve our manuscript. Below we have copied the text of those suggestions and added our response in italics (with red). We have incorporated manuscript changes based on almost all of these comments. Where we have given line numbers below, they refer to the version of the MS Word document without tracked changes.

Regards,

Yongjin Cho and co-authors

----------------------------------------------------

Response to comments: applsci-1413078

Reviewer #2

Comments:

Congratulations!

Great work done with very difficult subject of vision quality control system. Parac is very implementation-oriented, patent-pending, for practical use. No methodological reservations, a well-prepared review of the literature, a correctly described methodology of the work, clearly presented results and conclusions.

Response 1: Thank you for the great review and the good suggestions to improve our manuscript. We effort to revise the manuscript as you advised.

Point 1:

Please expand the descriptions of the algorithm revision systems from the literature review from the points of biography 3-10. Please specify their description.

Response 1: There is not enough page to describe the review of the algorithm from the literature. As you may know, the algorithm is well-described in the textbook, paper, and so on. You can find a description of the algorithm from the cited literature.  Instead of the additional descriptions of the algorithm, we added sentences for applications (ex, the machine or system that the algorithm was applied) in the revised manuscript as you advised (line 46-53).

Point 2:

Appearing abbreviations like RGB, SURF, SIFT ... - should be explained immediately after the first appearance of the abbreviation.

Response 2: We explained immediately after the first appearance of the abbreviation (line 44, 60, 103-108, line 246) in the revised manuscript as you advised.

Point 3:

Figure 3a - Improve photo quality

Response 3: We improved the photo quality (Figure 3a, page 5) in the revised manuscript as you advised.

Point 4:

What is the letter f in Formula 1?  What is this Formula, what is its source?

Same with Formula 2, 3 ...Explain all symbols in the formulas, e.g. 6, 7, 8 ...

Generally, Forlumas are unexplained by the Authors.

Response 4: We revised sentences and cited authors after Formula (line 201, 228-229, line 287-288, 297-300, 314-316, 339-340) in the revised manuscript as you advised.

Point 5:

Figure 11. Why are the sucker and node not sharp?

Response 5: We changed photo (Figure 11, page 13) in the revised manuscript as you advised

Point 6:

Consider whether in Figure 12 all photos are necessary.

Response 6:  Thank you for the good suggestion. However, we wish to keep Figure 12 in the manuscript. Because the other researcher who related to this topic may want the information of the visible spectrum image at each spectrum.

Point 7:

What do they contribute, how do they improve the quality of the manuscript?

I advise you to think about the necessity of a native speaker of the manuscript.

Response 7: We revised the manuscript by native speaker in the revised manuscript as you advised.
